# Factors associated with Macrosomia in public hospitals of Mekelle City, Northern Ethiopia: A multi-center study

Gebretinsae Alem Hagos[1], Mekuria Kassa Nerea[1], Edris Abdu Debesay[2], Mengistu Hagazi Tequare[1], Hiluf Ebuy Abraha[1,3], Yohannes Tesfay Abebe[1], Mohamedawel Mohamedniguss Ebrahim[1]*

1 Mekelle University, College of Health Sciences, Mekelle, Ethiopia, 2 Lekatit 11 Primary Hospital, Mekelle, Ethiopia, 3 Department of Epidemiology and Biostatistics, Arnold School of Public Health, University of South Carolina, Columbia, South Carolina, United States of America

* mohamedawel@gmail.com

## Abstract

### Background

Macrosomia, defined as a birth weight ≥4,000 grams, is linked to increased risks of cesarean delivery, postpartum hemorrhage, and neonatal morbidity, posing significant challenges in resource-limited regions like Northern Ethiopia., However, in Northern Ethiopia, evidence about the factors that lead to macrosomia is scarce. This study aimed to assess the factors associated with delivery of a macrosomic baby in public hospitals in Mekelle city, Northern Ethiopia.

### Method

A hospital-based, unmatched case control study design was implemented on 276 samples (184 controls and 92 cases) among newborns delivered in public hospitals of Mekelle City from February 21, 2020, to June 20, 2020. Cases and controls were selected using consecutive and systematic random sampling techniques, respectively. Data were collected using a structured questionnaire and a checklist and analyzed using SPSS. After describing the variables, bivariate and multivariable logistic regression models were employed to see the association between the factors and macrosomia. P < 0.05 was used to determine the statistical significance.

### Result

The multivariable model identified five factors that are significantly associated with macrosomia. As the age of the mother increases by one year, the odds of macrosomia were found to increase by 12% (AOR = 1.12, 95% CI: 1.02 to 1.23). As well, for each one-kilogram weight gain during pregnancy, the odds of macrosomia increased by 36% (AOR = 1.36, 95% CI: 1.12 to 1.65). And, one unit increment in body mass

**Data availability statement:** All relevant data are within the manuscript and its Supporting Information files.

**Funding:** The author(s) received no specific funding for this work.

**Competing interests:** The authors have declared that no competing interests exist.

**Abbreviations:** ACSH, Ayder Comprehensive Specialized Hospital, AOR, Adjusted Odds Ratio, BMI, Body Mass Index, COR, Crude Odds Ratio, MGH, Mekelle General Hospital, QGH, Quiha General Hospital, SD, Standard Deviation, VIF, Variance Inflation Factor.

index was found to increase the odds of macrosomia by 26% (AOR = 1.26, 95% CI: 1.06 to 1.50). Likewise, the male sex of the newborn and history of macrosomia were found to increase the chance of macrosomia by 2.66 (AOR = 2.66, 95% CI: 1.08 to 6.56) and 3.59-fold (AOR = 3.59, 95% CI: 1.62 to 7.95), respectively.

## Conclusion

In this study, the risk of macrosomia was found to be higher for male newborns, older mothers, greater weight gain during pregnancy, higher body mass index, and mothers with a history of macrosomia. This study has found both modifiable and non-modifiable risk factors of macrosomia. Policymakers should integrate preconception counselling and health education about the modifiable risk factors of macrosomia, such as weight gain during pregnancy. Big emphasis should be given to the male fetus and pregnant mothers with advanced age and history of macrosomia.

## Introduction

A macrosomic infant is an infant born with a birth weight greater than or equal to 4000 grams [1]. The incidence of fetal macrosomia varies from 1 to 20% of all deliveries. Evidence shows that pregnancies with macrosomic infants are associated with an increased risk of complications for both neonates and their mothers [2,3]. Maternal complications include postpartum hemorrhage, chorioamnionitis, prolonged labor, caesarean delivery, perineal trauma, and prolonged hospital stay. And, neonatal complications include shoulder dystocia, brachial plexus injury, skeletal injury, meconium aspiration, perinatal asphyxia, hypoglycemia, hypocalcemia, clavicular fracture, respiratory distress, low Apgar score, and death [2–4].

Birth weight is an important indicator in the prediction of short- and long-term health outcomes. Based on birth weight, the World Health Organization (WHO) classifies infants as Low Birth Weight (LBW) (<2500 g), Normal birth weight (2500–3999g), and Macrosomia (≥4000 g) [3].

Fetal growth can be considered to be the result of an interaction between the genetic potential for growth and the environmental influences. Insulin is the most important hormone in the regulation of fetal growth which is confirmed by fetal hyperinsulinemia in macrosomic syndrome such as maternal diabetes [5].

A diagnosis of fetal macrosomia can be made only by measuring birth weight after delivery; therefore, the condition is confirmed only retrospectively, after the delivery of the neonate [6].

Currently, the rates of macrosomic babies have been increasing in both developed and developing countries. Worldwide, there is a slight difference in the prevalence of macrosomia and this is due to the difference in the population studied. Globally, macrosomia affects 3–15% of all pregnancies. In developed world the magnitude of macrosomia is ranging from 5 to 20% of all births. More recent estimates indicate that macrosomia occurs in approximately 8% of all U.S.A live births [7,8]. In Asia, research done in China Beijing the incidence of macrosomia is found 7.1%.

The prevalence of macrosomia was assessed to be 21.2% among north Norwegian women, where as in Belgium 8.63% [9–11].

In Africa, studies in Ghana the prevalence of macrosomia is 10.5%, in the last two decades in Morocco ranges from 5.64% to 14.37% and the prevalence in Tanzania is found 2.3% [12–15]. Studies from Nigeria, Abuja showed the incidence to be 7.8 while in Cameroon the prevalence was 6.4% [16,17]. In Zambia the prevalence is found 2.5% where as in Niger it is found that 14.7% [18,19]. In Ethiopia a study in Hawassa showed the prevalence of macrosomia is 11.86% [7]. In Tigray Region, reported rates range from 7.5% to 19.1%, with higher prevalence documented one of the highest rates at 19.1%, underscoring potential geographic disparities in risk factor distribution [3,20]

According to study neonatal mortally in Ethiopia has remained stable from 29 to 30 per 1000 from 2016 to 2019 respectively [21]. Even though the country was doing great in achieving MDGs, neonatal and maternal mortality is still among the highest in the globe [22]. One of the factors contributing for both neonatal and maternal morbidity and mortality is macrosomia and its complication. Exploring the factors associated with macrosomia would contribute for achieving the SDG3(-neonatal mortality <12/1000 live births by 2030) [23].

The cause of fetal macrosomia is complicated and affected by a mixture of factors from the mother, fetus, and environment. Important risk elements for macrosomia in Ethiopia match worldwide patterns where obesity in mothers, too much weight gain during pregnancy, and gestational diabetes mellitus (GDM) are major contributors [20]. Further to this are indicators like high BMI before getting pregnant, carrying the baby beyond term (more than 42 weeks), and if the unborn child female; these also play roles suggesting both biological aspects as well as behavior that interacts [3]. Strikingly, research in Northwestern Ethiopia emphasized adjustable factors like insufficient prenatal follow-up, lack of physical activity during pregnancy and poor management of weight gain during gestation. All these present feasible targets for health intervention at the public level [24].

This research speaks about the issue of macrosomia in Tigray's public health system. Main focus is on Ayder Comprehensive Specialized Hospital, Mekelle General Hospital and Quiha General Hospitals that serve a varied economical demographic from all parts of the Tigray Region. Studies done before [20] usually looked at births occurring at private clinics which might not apply to these public hospitals – there are different types of patients with different income levels, nutritional conditions and clinical practices. Importantly, previous studies like [3,20] faced limitations as they were cross-sectional. This posed issues for determining cause-effect relations or making general predictions about wider populations using those findings. The same regional studies [25] did in Northern Ethiopia may not mirror the specific issues that Tigray face, highlighting a call for research within local limits [24]. Also, previous investigations [3,20] left out important variables like residency (an essential socio-demographic element), gestational diabetes, history of diabetes and family record of high blood pressure gaps this investigation is dealing with.

Moreover, this research addresses key contextual, methodological, and unique variable-related gaps in macrosomia studies. Firstly, it places primary focus on the public healthcare environment at Ayder Hospital where economic disparities and a lack of specialized care increase risks. Secondly, it improves statistical precision beyond earlier researches by providing practical knowledge for prenatal tactics like specific weight management and screening. As case-control research, it gives more solid base for upcoming group studies to determine causality – a limitation of previous cross-sectional patterns. At last, it matches with Ethiopia's health goals by creating proof to lessen avoidable mother and newborn problems in settings with less resources.

This research, different from past studies [3,20,24], incorporates factors like living place, diabetes during pregnancy and family health histories in risk models. These elements were not thoroughly looked at before in comparable circumstances. The aim of this research is to help health workers and policy makers with context-specific strategies such as better screening for diabetes and hypertension among high-risk groups. This could improve outcomes in Northern Ethiopia.

The results of this research will add to the small amount of information on macrosomia in areas with fewer resources and give suggestions based on evidence for bettering mother and newborn health conditions in Northern Ethiopia. By

tackling particular risk factors and medical problems in this area, the research gives a detailed comprehension of macrosomia which enhances current global and city-focused studies.

## Methods

### Study design, area, and period

A hospital- based unmatched case-control study was conducted in the public hospitals of Mekelle from February 21, 2020 to June 20, 2020. Mekelle is a city found in Tigray Regional state, Ethiopia, and located around 781 kilometers north to Addis Ababa. In Mekelle, there are five public hospitals and eight health centers [20].

For this study, we selected three public hospitals (i.e., Ayder Comprehensive Specialized Hospital (ACSH), Mekelle General Hospital (MGH), and Quiha General Hospital (QGH)) by lottery method. ACSH has a capacity of 500 beds and it is the only tertiary hospital for a population of 8.5 million in Tigray, Afar, and Southeast Amhara. MGH is the oldest regional hospital in the region and established in 1962. QGH was established in 1985 by Italian cooperation and its current capacity is around 50 beds. ACSH and MGH each entertain more than 5000 maternal deliveries annually. However, QGH manages around 700 deliveries annually.

### Study population

The study population for cases were macrosomic newborns delivered in selected hospitals during the data collection period. And, for controls, they were newborns with normal birth weight who were delivered in selected hospitals during the data collection period.

### Eligibility criteria

The research included of single live births that occurred at 37 weeks or beyond in the gestation period (term births). The weight of these newborn babies must be evaluated within a day after they are born using a specific scale, with a bar set at least 4000 grams. There is also an obligation for mothers and their newly-born to live in the Tigray area. To participate, it is necessary for the mother to agree and permit access to her medical history and data of her newborn. The childbirth must take place in health institutions such as hospitals or health centers, or need trained healthcare workers present if happening at community locations, so that we can collect reliable information. Also note: only births that happened during this study period are considered.

The research does not include multiple pregnancies, still birth, preterm births and also the cases of birth defects. If there are mothers or newborns for whom we lack critical information like weight at birth, gestational age or medical history of mother then they will also be left out to keep accuracy in our analysis high. People who do not live in the Tigray region, maternal psychiatric problems were used as exclusion criteria.

### Variables

The dependent variable was birthweight categorized as normal and macrosomia. ***Socio-demographic characteristics*** (age, marital status, job, level of education, residence, religion, monthly family income, family size, physical exercise), ***obstetric factors*** (parity, length of gestation, sex of the newborn, birth weight of the last child, gestational diabetes mellitus, inter pregnancy interval, type of pregnancy), ***Anthropometric measurements*** (maternal weight, maternal height, weight of the newborn, body mass index (BMI)) and ***chronic illnesses*** were considered as independent variables.

### Operational definitions

**Macrosomia**- is the weight of the newborn greater than or equal 4000g [1,2,26].
**Maternal weight** -the weight of the mother measured in the late first trimester [15].

**Body mass index (BMI)** -the weight of the mother measured in the late first trimister divided by the sequere of the height of the mother in meters [16].

**Total gestational weight gain**-the deference of maternal weight in kilogram in the first trimester and maternal weight during delivery in kilogram [18].

## Sample size determination

The sample size was calculated using the double population proportion formula for the unmatched case control study by taking the maternal weight during delivery ≥ 90 kg as the exposure variable. Proportion of cases and controls with maternal weight ≥90 kg was taken from previous studies and it was 29.1% and 14.5%, respectively [15]. Maternal weight was chosen because it gave the maximum sample size compared to other reviewed predictors of macrosomia. Considering a confidence interval of 95%, a power of 80%, and the ratio of controls to cases as 2:1, the minimum total sample size required was 252. By adding a 10% nonresponse rate, the final total sample size was 276 (184 controls and 92 cases). Software used for sample size calculation was EpiInfo version 7.2.

## Sampling procedures and techniques

The sample size was allocated to the selected hospitals proportionally based on the number of newborns delivered in the labor ward of each hospital in the preceding two months, which was taken from November 1-December 31/2019. Accordingly, in a couple of months, there were 1032 normal birth weight (480 in ACSH, 502 in MGH, and 50 in QGH) and 100 macrosomic (45 in ACSH, 47 in MGH and 8 in QGH) babies. Cases were recruited consecutively until the calculated sample size was attained while controls were selected using a systematic random sampling technique by using the formula:

$$K = \frac{N(\textit{Total number of controls})}{n\ (\textit{required number of controls})} = \frac{1032}{184} = 6.$$

As a result, controls were selected every sixth newborn with normal birth weight. The distribution of recruited cases was 42, 43, and 7 for ACSH, MGH, and QGH, respectively. Two controls were recruited for each case from the corresponding hospital.

## Data collection tools and procedures

Structured interviewer administered questionnaires and checklists were used to collect data related to socioeconomic and demographic factors, obstetric factors, anthropometrical measurements, and health factors. The structured questionnaire was used to collect data from the mother through direct interviews, and the checklist was used to extract data from the medical records of both the mother and the newborn [3,7,15,20,24,26–28]. The data were collected by four trained midwives from February 21, 2020 to June 20, 2020.

## Data quality control

The questionnaire was prepared in English and then was translated to Tigrigna which is the local language of the study area. Then it was translated back to English by language experts to see its consistency. The questionnaire was reviewed by senior researchers and feedback was incorporated accordingly. One day training was given to the interviewers concerning the questionnaire, interviewing technique, purpose of the study, purpose of maintaining the subject's privacy, discipline, and keeping confidentiality. Case ascertainment was done by looking the medical record of the mother. The data was checked for coding errors, consistency, and completeness.

## Data management and analysis

The collected data were coded, entered, cleaned, and analyzed using SPSS version 23. Categorical variables were described using frequencies and percentages. After graphically and statistically checking for normality of distribution, continuous variables

were described using an appropriate combination of measures of central tendency and dispersion. Relationship between categorical variables and birth weight was evaluated using chi-square test of association. For continuous variables, a mean comparison between macrosomic and normal birth weight babies was made using either Independent-Samples t-test or Welch t-test. The choice of t-test was determined by Levene's test of equality of variance. Independent-Samples t-test was used during insignificant Levene's test result. On the contrary, Welch t-test was used during significant Levene's test results.

Binary logistic regression model was used to see the association between macrosomia and the independent variables. Those independent variables with $p < .25$ during bivariate analysis were exported to a multivariable logistic regression model. Goodness of fit and multicollinearity were checked using Hosmer-Lemeshow test and Variance Inflation Factor (*VIF*), respectively. Statistical significance was declared at $p < .05$.

### Ethical consideration

The Mekelle University-College of Health Sciences' research ethics review board (IRB) granted ethical approval with ethical clearance number: ERC 1539/2020. Official approval was acquired by ACSH's clinical director. After being properly informed about the study and signing a formal consent agreement, study participants voluntarily participated. Additionally, the participants were made aware of their freedom to leave the study at any time. Confidentiality was maintained during the entire study period, and all data was anonymously recorded.

## Results

A total of 276 participants were included in this study, of which 92 were cases and 184 were controls with 100% response rate.

### Socio-demographic characteristics

Our study indicated that the mean (SD) age of mothers among cases and controls was 33.9(4.7) and 27.5(4.0) years, respectively. Male newborns accounted for 78.3% of cases and 33.7% of controls. Regarding family size, 93.7% cases and 64.1% of controls were from a family with three and above members (**Table 1**).

### Obstetrics, Anthropometric, and Health-related factors

More than one-third (34.8%) of the cases and 6(3.3%) of the controls were grand multipara. The mean of first trimester BMI was $27.06 \pm 2.19$ kgs/m2 and $24.05 \pm 2.19$ kgs/m2 for case and controls, respectively. The mean total gestational weight gain was $13.22 \pm 2.28$ kilograms for cases and $10.99 \pm 1.75$ kilograms for controls. The mean birth interval for the cases and controls was $2.38 \pm 1.00$ and $1.72 \pm 1.14$ years, respectively. More than half (52.2%) of the cases and the majority (85.3%) of the controls were born before forty weeks of gestation. Sixty-five (70.7%) of the cases and 24 (13.0%) of the controls had a prior macrosomic baby. Thirty-three (35.9%) of the cases and 14 (7.6%) of the controls were diagnosed with gestational diabetes (**Table 2**).

### Factors associated with Macrosomia

A multivariable binary logistic regression was conducted to determine which independent variables (age of the respondents, weight gain, body mass index, birth interval, family history of diabetes mellitus, history of macrosomic infant delivery, maternal diabetes, gestational age, sex of the newborn, and gestational diabetes mellitus) were factors significantly associated with macrosomia. Hosmer and Lemeshow test results confirmed that the model was a good fit for the data, $X^2$ ($df = 8$, $N = 276$) = 12.15, $p = .145$.

Among the ten independent variables, five (age of the respondents, weight gain, body mass index, history of macrosomic infant delivery, and sex of the newborn) of them were statistically significant factors associated with macrosomia.

**Table 1. Sociodemographic characteristics of mothers and newborns from the selected public hospitals in Mekelle city, Tigray, Ethiopia,2020, N = 276.**

| Variable | Group of respondents | | | | | P value[a] |
|---|---|---|---|---|---|---|
| | Cases(n=92) | | Controls (n=184) | | | |
| Category | n | Column % | n | Column % | | |
| Age in years (Mean ±SD) | 33.9 ± 4.7 | | 27.5 ± 4.0 | | | <0.001[b] |
| Residence | | | | | | 0.553 |
| Urban | 85 | 92.4 | 166 | 90.2 | | |
| Rural | 7 | 7.6 | 18 | 9.8 | | |
| Sex of the newborn | | | | | | <0.001 |
| Male | 72 | 78.3 | 62 | 33.7 | | |
| Females | 20 | 21.7 | 122 | 66.3 | | |
| Educational status | | | | | | 0.022 |
| No formal education | 2 | 2.2 | 4 | 2.2 | | |
| Primary school | 40 | 43.5 | 105 | 57.1 | | |
| Secondary school | 23 | 25.0 | 49 | 26.6 | | |
| More than secondary school | 27 | 29.3 | 26 | 14.1 | | |
| Family size | | | | | | <0.001 |
| < 3 | 6 | 6.5 | 66 | 35.9 | | |
| ≥ 3 | 86 | 93.5 | 118 | 64.1 | | |

[a]A relationship between group and other variables was evaluated using chi-square test of association.

[b]Comparison of age between cases and controls was made using an Independent-Samples T test

Accordingly, as age of the mother increases by one year, the odds of macrosomia increase by 12% (AOR = 1.12, p = .021). For each one-kilogram weight gain during pregnancy, the odds of macrosomia increased by 36% (AOR = 1.36, p = .002). Similarly, for one unit increment in BMI, the odds of macrosomia increased by 26% (AOR = 1.26, p = .008). Likewise, the likelihood of delivering a macrosomic baby was 2.66 times higher in mothers with a history of macrosomia (AOR = 2.66, p = .034). Lastly, male newborns had more than 3 times higher odds of being a macrosomic baby (AOR = 3.59, p = .002) (Table 3).

## Discussion

In this study, an attempt was made to determine the associated factors of macrosomia among mothers who gave birth in public hospitals of Mekelle, Northern Ethiopia. Results of this study revealed that maternal age, total weight gain during pregnancy, early pregnancy BMI, prior macrosomic delivery, and male sex of the new born were found to be significant associated with macrosomia.

Our study has shown that the probability of macrosomia increases with the increment of maternal age. This has been demonstrated by studies done in China, Zambia, and Democratic Republic of Congo [18,27,29]. For example, in China's study, age was significantly higher in mothers who delivered a macrosomic baby (p < 0.001) and the odds of macrosomia increased by 4% for each one-year increment in maternal age [29]. This might be due to the fact that increasing maternal age may affect maternal metabolism by making the body less sensitive to insulin, thereby increasing the growth velocity of the fetus through the transfer of more nutrients to the fetus [15].

In our study, the odds of macrosomia were significantly higher in mothers who gained more weight during pregnancy. This finding is consistent with studies done in Ethiopia, Nigeria, Colombia, and Cameroon [16,17,20,24,25]. According to the studies of Nigeria and Cameroon, the likelihood of macrosomia was ten times higher in mothers with a total pregnancy weight gain of 15 kg and more than their counterparts [16,17]. This could be due to the increased accumulation of cellular water, fat, and

**Table 2.** Obstetric, anthropometrical, and health-related characteristics among newborn babies delivered in selected public hospitals in Mekelle city, Tigray, Ethiopia, 2020, N = 276.

| Variables | Group of respondents | | | | | P value[a] |
|---|---|---|---|---|---|---|
| | Cases (n = 92) | | Controls (n = 184) | | | |
| Categories | n | Column % | n | Column % | | |
| BMI (Mean ± SD), Kg/m$^2$ | 27.06 ± 2.19 | | 24.05 ± 2.19 | | | <0.001[b] |
| Weight gain (Mean ± SD), Kg | 13.22 ± 2.28 | | 10.99 ± 1.75 | | | <0.001[c] |
| Birth interval (Mean ± SD), years | 2.38 ± 1.00 | | 1.72 ± 1.14 | | | <0.001[b] |
| Gestational age(weeks) | | | | | | <0.001 |
| < 40 | 48 | 52.2 | 157 | 85.3 | | |
| ≥ 40 | 44 | 47.8 | 27 | 14.7 | | |
| P arity | | | | | | <0.001 |
| 1 | 6 | 6.5 | 62 | 33.7 | | |
| 2-4 | 54 | 58.7 | 116 | 63.0 | | |
| ≥ 5 | 32 | 34.8 | 6 | 3.3 | | |
| Prior macrosomic delivery | | | | | | <0.001 |
| Yes | 65 | 70.7 | 24 | 13.0 | | |
| No | 27 | 29.3 | 160 | 87.0 | | |
| Gestational diabetes | | | | | | <0.001 |
| Yes | 33 | 35.9 | 14 | 7.6 | | |
| No | 59 | 64.1 | 170 | 92.4 | | |
| Diabetes | | | | | | <0.001 |
| Yes | 29 | 31.5 | 14 | 7.6 | | |
| No | 63 | 68.5 | 170 | 92.4 | | |
| Family history of diabetes | | | | | | <0.001 |
| Yes | 36 | 39.1 | 32 | 17.4 | | |
| No | 56 | 60.9 | 152 | 82.6 | | |
| Family history of hypertension | | | | | | <0.001 |
| Yes | 37 | 40.2 | 17 | 9.2 | | |
| No | 55 | 59.8 | 167 | 90.8 | | |
| Hypertension | | | | | | <0.001 |
| Yes | 24 | 26.1 | 2 | 1.1 | | |
| No | 68 | 73.9 | 182 | 98.9 | | |

[a]A relationship between group and other variables was evaluated using chi-square test of association.

[b]Comparison of BMI and birth interval between cases and controls was made using an Independent-samples t test

[c]Comparison of weight gain between cases and controls was made using Welch t-test

protein during pregnancy that in turn allows the passage of excess free fat mass through the placenta to the fetus leading to escalation of fetal weight or macrosomia [30]. These connections are strong, but variations in the impact sizes from different studies might come from differences in population characteristics, how "excessive" weight gain is defined or factors related to diet and lifestyle depending on the area. For example, a Mekelle research pointed out pre-pregnancy overweight and obesity as an influencing factor [20] which our analysis did not identify indicating that conditions existing before pregnancy may increase the effects of gaining weight during pregnancy. Differences in availability of prenatal care or nutritional condition could also play a part. These results highlight that the age and weight gain of mothers are important factors for macrosomia risk.

This study has revealed a significant relationship between macrosomia and early pregnancy body mass index (BMI). This was in line with studies from China and Zambia [18,29]. These studies found that mothers who were

**Table 3. Factors associated with macrosomia among newborn babies in selected hospitals of Mekelle, Tigray, Ethiopia, 2020, N = 276.**

| Variables | COR (95% CI) | AOR (95% CI) |
|---|---|---|
| Age of mother in years | 1.36 [1.27, 1.46] | 1.12 [1.02, 1.23]* |
| Weight gain in Kg | 1.74 [1.50, 2.02] | 1.36 [1.12, 1.65]** |
| BMI (Kg/m²) | 1.76 [1.52, 2.03] | 1.26 [1.06, 1.49]** |
| Birth Interval, years | 1.76 [1.37, 2.27] | 1.15 [0.78, 1.71] |
| Family history of diabetes | | |
| Yes | 3.05 [1.73, 5.38] | 1.08 [0.45, 2.57] |
| No | 1 | 1 |
| Previous history of macrosomia | | |
| Yes | 16.05 [8.63, 29.86] | 2.66 [1.08, 6.56]* |
| No | 1 | 1 |
| Maternal diabetes | | |
| Yes | 5.59 [2.78, 11.26] | 1.07 [0.37, 3.04] |
| No | 1 | 1 |
| Gestational age | | |
| < 40 weeks | 1 | 1 |
| ≥ 40weeks | 5.33 [2.99, 9.50] | 0.96 [0.39, 2.36] |
| Gestational diabetes | | |
| Yes | 6.79 [3.40, 13.56] | 2.24 [0.82, 6.16] |
| No | 1 | 1 |
| Sex of the newborn | | |
| Male | 7.08 [3.96, 12.68] | 3.59 [1.63, 7.91]** |
| Female | 1 | 1 |
| Residence | | |
| Urban | 1 | |
| Rural | 0.76 [0.31, 1.89] | |
| Educational status of the mother | | |
| No formal education | 1 | |
| Primary school | 0.76 [0.13, 4.32] | |
| Secondary school | 0.94 [0.16, 5.50] | |
| More than secondary school | 2.08 [0.35, 12.33] | |
| Type of pregnancy | | |
| Unplanned and unwanted | 1 | |
| Wanted but unplanned | 0.39 [0.14, 1.12] | |
| Wanted and planned | 0.59 [0.23, 1.52] | |

*p<0.05

**p<0.01

***p<0.001

either overweight or obese during early pregnancy had 1.5 to 2.8 times higher odds of delivering a macrosomic baby. This is because overweight or obesity increases insulin resistance resulting in higher fetal glucose and insulin levels. In addition, placental lipases metabolize maternal serum triglycerides, thus allowing free fatty acids to be transferred in excess to the growing fetus [2]. These bodily processes emphasize how directly the metabolic condition of a mother can affect her unborn child's health outcomes. Thus, we realize more how

important it is for women to maintain healthy Body Mass Index (BMI) before their pregnancy because it may present certain risks otherwise. To avoid high weight gain during pregnancy and lessen the risk of macrosomia, it is important to do early checks for raised BMI and take steps such as nutrition advice. Future research work should look at how placental function works along with glucose metabolism so we can have better knowledge about these processes. This will also help in making our preventative methods more accurate for different kinds of people.

According to our study, history of macrosomia was significantly related with macrosomia in current pregnancy by increasing its risk. Congruently, a cross-sectional study conducted in Southern Ethiopia among women who gave birth in a public health institution reported a remarkably higher chance of having a macrosomic baby in mothers with a history of macrosomia [7]. Likewise, studies of Zambia, Nigeria, Cameroon, and Democratic Republic of Congo support this relationship [15–17, 26]. Recurrence of fetal macrosomia may be due to greater maternal BMI at the time of conception, excessive weight gain between pregnancies as well as weight gain during pregnancy [2,15,30].

Lastly, we have found a significant association between sex of the newborn and macrosomia. Accordingly, male newborns had about 3.6 times higher odds of being macrosomic after adjusting for the rest nine variables in the final multivariable model. This was in tune with studies conducted in Southern Ethiopia, Zambia, Italy, China, and Bosnia and Herzegovina [7,18,29,31,32]. This may attributed to the growth patterns specific to sex which may be caused by differences in placental or hormonal factors [20,24].

## Limitations of the study

Due to the limited availability of pregestational weight, we calculated BMI in early pregnancy. This might have affected our BMI to some extent. The fact that our study was a case-control study could have introduced a recall bias. Moreover, this research was carried out only in Mekelle city, the results might not reflect other parts of Ethiopia or wider Tigray region. Variances in socioeconomic, cultural and healthcare aspects throughout different regions could affect the occurrence and factors behind macrosomia. It's suggested to take care when applying these findings generally.

## Conclusions

Maternal age, weight gain during pregnancy, early pregnancy BMI, history of macrosomic delivery, and male sex of the new born were found to be significantly associated with macrosomia. Policy makers should integrate preconception counselling and health education about the modifiable risk factors of macrosomia such as weight gain during pregnancy. Great emphasis should be given to the male fetus and mothers with advanced age and history of macrosomia.

## Supporting information

**S1 File. SAV.** SPSS dataset. This file contains the dataset used in this study in SPSS file format.
(SAV)

## Acknowledgments

We would like to forward our gratitude to all people who helped from the preparation of the proposal up to the final manuscript submission.

## Author contributions

**Conceptualization:** Gebretinsae Alem Hagos, Mekuria Kassa Nerea, Mengistu Hagazi Tequare, Yohannes Tesfay Abebe.
**Data curation:** Mohamedawel Mohamedniguss Ebrahim, Gebretinsae Alem Hagos.

**Formal analysis:** Mohamedawel Mohamedniguss Ebrahim, Gebretinsae Alem Hagos, Mengistu Hagazi Tequare, Hiluf Ebuy Abraha, Yohannes Tesfay Abebe.

**Investigation:** Gebretinsae Alem Hagos, Yohannes Tesfay Abebe.

**Methodology:** Mohamedawel Mohamedniguss Ebrahim, Gebretinsae Alem Hagos, Mekuria Kassa Nerea, Mengistu Hagazi Tequare, Hiluf Ebuy Abraha, Yohannes Tesfay Abebe.

**Project administration:** Gebretinsae Alem Hagos.

**Resources:** Gebretinsae Alem Hagos.

**Software:** Mohamedawel Mohamedniguss Ebrahim, Gebretinsae Alem Hagos, Mengistu Hagazi Tequare, Hiluf Ebuy Abraha.

**Supervision:** Gebretinsae Alem Hagos, Mekuria Kassa Nerea, Yohannes Tesfay Abebe.

**Validation:** Gebretinsae Alem Hagos, Mekuria Kassa Nerea, Edris Abdu Debesay, Yohannes Tesfay Abebe.

**Visualization:** Mohamedawel Mohamedniguss Ebrahim, Edris Abdu Debesay, Yohannes Tesfay Abebe.

**Writing – original draft:** Mohamedawel Mohamedniguss Ebrahim, Gebretinsae Alem Hagos, Mekuria Kassa Nerea, Edris Abdu Debesay, Mengistu Hagazi Tequare, Hiluf Ebuy Abraha, Yohannes Tesfay Abebe.

**Writing – review & editing:** Mohamedawel Mohamedniguss Ebrahim, Gebretinsae Alem Hagos, Mekuria Kassa Nerea, Edris Abdu Debesay, Mengistu Hagazi Tequare, Hiluf Ebuy Abraha, Yohannes Tesfay Abebe.

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
