## [Decision Letter · Decision Letter 0]

4 Feb 2025

PONE-D-24-60087Factors Associated with Macrosomia in Public Hospitals of Mekelle City, Northern Ethiopia: A Multi-Center StudyPLOS ONE

Dear Dr. Ebrahim,

Thank you for submitting your manuscript to PLOS ONE. After careful consideration, we feel that it has merit but does not fully meet PLOS ONE’s publication criteria as it currently stands. Therefore, we invite you to submit a revised version of the manuscript that addresses the points raised during the review process.

We look forward to receiving your revised manuscript.

Kind regards,

Tamirat Getachew

Academic Editor

PLOS ONE

Journal Requirements:

Reviewers' comments:

Reviewer's Responses to Questions

**Comments to the Author**

1. Is the manuscript technically sound, and do the data support the conclusions?

Reviewer #1: Partly

Reviewer #2: Yes

2. Has the statistical analysis been performed appropriately and rigorously? 

Reviewer #1: No

Reviewer #2: Yes

3. Have the authors made all data underlying the findings in their manuscript fully available?

Reviewer #1: No

Reviewer #2: Yes

4. Is the manuscript presented in an intelligible fashion and written in standard English?

Reviewer #1: No

Reviewer #2: No

5. Review Comments to the Author

Reviewer #1: Review report

It is a great pleasure for me to review the manuscript of this demanding public health problem conducted in the Tigray region of northern Ethiopia. However, this manuscript is not suitable for publication in its current standing, as it needs major revision in the following sections and I hope the authors will be benefitted from this feedback to further improve the manuscript for scientific publication.

Title

The title requires revisions, as the authors have indicated it follows a case-control design. Therefore, the phrase "factors associated" is not the most suitable epidemiological terminology for this type of study design. Furthermore, the title should be revised to accurately reflect the study design and ensure appropriate epidemiological terms.

Abstract and introduction section.

The background/introduction section requires enhancement as it currently lacks depth and does not clearly articulate the problem at hand. It is important to establish a strong rationale for conducting this research. What unique contributions does this study bring compared to previous work, and what justifications can be provided? Although the background states that there is limited knowledge about the factors leading to macrosomia in Northern Ethiopia, it is essential to acknowledge that existing studies have already documented these factors. Recognizing this literature will add valuable context to the research.

e.g.,

1. https://bmcpediatr.biomedcentral.com/articles/10.1186/s12887-017-0901-1.

2. https://pubmed.ncbi.nlm.nih.gov/31262264/.

3. https://journals.sagepub.com/doi/10.1177/03000605221132028?icid=int.sj-full-text.similar-articles.6.

Objectives: This section needs revision based on the comments given above.

Results: This section needs significant revision, please write the adjusted measures of effect (AOR with its 95% CI). Likewise, try to make the interpretation as meaningful as possible. How do you interpret risk and preventive odds ratio? As the age of the mother increases by one year, the odds of macrosomia were found to increase by 12%?? The odds of macrosomia increased by 36% and 26%?? Please revise the interpretation by considering these recommendations.

Study design, area, and Period

• The authors have focused exclusively on Mekelle City for their research, raising questions about the decision not to include other general hospitals in the region, such as those in Axum, Adigrat, Maichew, Abiyi Adi, Suhul Shire, and Kidiste Mariam, among others. This choice prompts a discussion about the factors influencing the selection of Mekelle city over the six or seven zones that make up the region. Understanding the rationale behind this focus could provide insight into the research objectives and outcomes.

• The previously conducted studies in the Tigray region were also in Mekelle, why this study is also in Mekelle without clear added value?

• Mekelle is 783 kilometers from Addis Ababa

• Put reference for the mentioned statements

Eligibility criteria

• The authors mentioned only the exclusion criteria, what about the inclusion criteria?

• Macrosomia- is the weight of the newborn greater than or equal 4000g [1,2,27]. Use only a single reference.

• predictors of macrosomia vs determinants vs factors associated, which one are the authors studying?

Data collection tools and procedures

• Put reference for the data collection tool, from which literatures did the authors use to develop the tool?

• The collected data were coded, entered, cleaned, and analyzed using SPSS version 23. Why the authors fail to use data entry software so that they can minimize data entry errors. Any explanation?

• In the data analysis plan, please mention that the crude and adjusted measures of effect (COR and AOR), how do the authors select candidate variables from bivariate to multivariable model? Did the authors include those variables which were statistically non-significant but clinically important in their model?

• Is it possible to calculate percent if the denominator is less than 100, e.g., the total number of cases is 92, which is <100. Is it appropriate say “Male newborns accounted for 78.3% of cases”?? Please mention only the numbers if the denominator is less than 100. Do not calculate percentages.

• What is the interpretation for the significant difference in the baseline sociodemographic characteristics of the cases and controls? Similarly, the cases and controls were not comparable based on their Obstetric, anthropometrical, and health-related characteristics? If they were not comparable from the beginning, do you think the findings are reliable and generalizable?

• Predictors of Macrosomia: This is not appropriate ate epidemiologic term for your study design. Please revise and present appropriately.

• No need to present the multicollinearity, the omnibus test of model coefficients was significant, Cox and Snell residuals, Nagelkerke’s R2 and the ROC curve analysis in the result section.

• However, the Hosmer and Lemeshow test results should be presented in the final regression table.

• The final regression table should be presented again containing the non-significant variables too. Additionally present the values of for the macrocosmic and normal weight babies cross tab.

• Additionally, present Asterix or any other symbol for the significant variables.

• The discussion section is very shallow and try to discuss in detail the possible justification for any similarity or differences in the findings, implication of the findings for community, patients, researchers, and policy makers.

• Try to compare studies conducted in low income set up and specifically Ethiopia and Tigray region.

• What were the strengths of the study?

• Pleas mention in the method that case ascertainment was done by looking the medical record of the mother.

• What efforts are made to control or minimize bias?

Reviewer #2: Title: Factors Associated with Macrosomia in Public Hospitals of Mekelle City, Northern

Ethiopia: A Multi-Center Study

Dears,

We express our sincere gratitude to the authors for their valuable research on this important public health issue. We also thank the editors for providing us with the opportunity to review this crucial work.

General Comments:

While the study provides valuable insights into macrosomia in Mekelle City, a more comprehensive review of existing literature on macrosomia in Ethiopia is crucial. The current discussion primarily compares with studies conducted in other countries (China, Nigeria, etc.). The authors should dedicate a section to discussing the findings of this study in the context of existing Ethiopian research on macrosomia. This would enhance the local relevance and impact of the study. A more thorough literature review on macrosomia in Ethiopia would allow for a better understanding of the unique challenges and risk factors specific to the Ethiopian context.

Specific Suggestions:

- Please format your abstract to include the following subheadings are Background, Methods, Results, and Conclusions. Therefore, please remove the Objectives subheading and include the text in one of the above sections.

- The authors should incorporate citations (reference) in some sentences e.g. introduction part paragraph 1 line 2, paragraph 2 lines 2-3.

- Authors should revise paragraphs one and two of the introduction to reduce redundancy. These two sentences convey essentially the same information. To reduce redundancy, the authors could combine the sentences

- In the introduction, the authors should include a dedicated paragraph summarizing key findings from relevant studies conducted in Ethiopia on the prevalence and risk factors of macrosomia

- Inconsistency in how birth weight is categorized in introduction and methods. If there is a valid reason for using two categories (normal and macrosomia) in the methods and analysis, clearly explain this rationale in the introduction or discussion.

- How did the authors screen for pre-existing and gestational diabetes?

- Result section---the sentences “Our study indicated that the mean (SD) age of cases and controls was 33.9(4.7) and 27.5(4.0) years. Maternal age or newborn age represent?

- The discussion section should be expanded to include a comparative analysis of the study's findings with those of other studies conducted in Ethiopia. This could highlight similarities, differences, and potential explanations for any observed discrepancies.

- The authors should discuss the implications of their findings for public health interventions in Ethiopia, considering the specific context and existing healthcare resources.

- Limitation of the study should be discussed, especially the control for preexisting and gestational diabetes which are the very important risk factors for macrosomia

- The authors should also discuss about the generalizability of the study results since the samples were only from mekelle city.

6. PLOS authors have the option to publish the peer review history of their article (what does this mean? ). If published, this will include your full peer review and any attached files.

**Do you want your identity to be public for this peer review?** For information about this choice, including consent withdrawal, please see our Privacy Policy .

Reviewer #1: **Yes: ** Zenawi Hagos Gufue

Reviewer #2: No

---

## [Author Response · Author response to Decision Letter 0]

11 Apr 2025

Factors Associated with Macrosomia in Public Hospitals of Mekelle City, Northern Ethiopia: A Multi-Center Study

A point-by-point response to reviewers

Dear Editors and Reviewers, we highly appreciate for your constructive comments, feedbacks, and suggestions which helped us improve the quality of our work. Please kindly find our point-by-point response to your precious comments and suggestions.

Reviewer #1:

Comment 1: [" The title requires revisions, as the authors have indicated it follows a case-control design. Therefore, the phrase "factors associated" is not the most suitable epidemiological terminology for this type of study design. Furthermore, the title should be revised to accurately reflect the study design and ensure appropriate epidemiological terms."]

Response: We thank the reviewer for this suggestion. Epidemiological Case control studies typical use factor associated to reflect correlation, not causation, unless longitudinal evidence exists.

Determinants often imply causation (e.g., a direct cause-effect relationship). Since cofounding variable may exist and causality cannot be confirmed in a case-control study, associated factor is more accurate and conservative

Comment 2: ["The Abstract and introduction section.

The background/introduction section requires enhancement as it currently lacks depth and does not clearly articulate the problem at hand. It is important to establish a strong rationale for conducting this research. What unique contributions does this study bring compared to previous work, and what justifications can be provided? Although the background states that there is limited knowledge about the factors leading to macrosomia in Northern Ethiopia, it is essential to acknowledge that existing studies have already documented these factors. Recognizing this literature will add valuable context to the research”.

Response: We agree with the reviewer that abstract and introduction section need enhancement. We have revised this section; we have added a paragraph to clarify the problem at hand and to establish a strong rational on introduction section. We have added and acknowledged the previous studies.

Comment 3: [" Objectives: This section needs revision based on the comments given above.

Response: We thank the reviewer for this suggestion. Epidemiological Case control studies typical use factor associated to reflect correlation, not causation, unless longitudinal evidence exists.

Comment 4: Results: This section needs significant revision, please write the adjusted measures of effect (AOR with its 95% CI). Likewise, try to make the interpretation as meaningful as possible. How do you interpret risk and preventive odds ratio? As the age of the mother increases by one year, the odds of macrosomia were found to increase by 12%?? The odds of macrosomia increased by 36% and 26%?? Please revise the interpretation by considering these recommendations.

Study design, area, and Period"]

Response: We thank the reviewer for this suggestion. We have revisited the statistical analysis and included additional details about the methods used in the revised manuscript, we have added AOR with it 95% CI).

Comment 4: [ " Study design, area, and Period

• The authors have focused exclusively on Mekelle City for their research, raising questions about the decision not to include other general hospitals in the region, such as those in Axum, Adigrat, Maichew, Abiyi Adi, Suhul Shire, and Kidiste Mariam, among others. This choice prompts a discussion about the factors influencing the selection of Mekelle city over the six or seven zones that make up the region. Understanding the rationale behind this focus could provide insight into the research objectives and outcomes.

• The previously conducted studies in the Tigray region were also in Mekelle, why this study is also in Mekelle without clear added value?

• Mekelle is 783 kilometers from Addis Ababa

• Put reference for the mentioned statements

Responses: Thanks for your comments we have add references on manuscript

NB: The previous study was conducted in private clinic but the current study was at public hospitals furthermore Ayder Hospital, a high-volume tertiary center serving socioeconomically diverse populations. While prior studies, such as [25], provided insights from private clinic births, their findings may lack generalizability to public hospitals like Ayder due to differences in patient demographics (e.g., income levels, nutritional status) and clinical practices. Critically, earlier clinical-based studies—including Tela et al. (2019)—were limited by their cross-sectional designs, which hinder causal inference and limit extrapolation to broader populations. Furthermore, previous work omitted key variables such as residency (a critical socio-demographic factor), gestational diabetes, diabetes history, and family history of hypertension—gaps this study addresses.

Comment: Eligibility criteria

• The authors mentioned only the exclusion criteria, what about the inclusion criteria?

• Macrosomia- is the weight of the newborn greater than or equal 4000g [1,2,27]. Use only a single reference.

• predictors of macrosomia vs determinants vs factors associated, which one are the authors studying?"]

Response: We thank the reviewer for this suggestion. We have added reference and Eligibility criteria specifically inclusion criteria, we choose single reference #1 for weight, and for distance of Mekelle city from Adis Abeba. Thanks for asking this question we have accepted this comments However, Determinates often implies causation (e.g., a direct cause-effect relationship). Since cofounding variable may exist and causality cannot be confirmed in a case-control study, associated factor is more accurate and conservative, thus we are studying Factor associated with macrosomia Among Newborns

Comments 5: “Data collection tools and procedures”

• Put reference for the data collection tool, from which literatures did the authors use to develop the tool?

• The collected data were coded, entered, cleaned, and analyzed using SPSS version 23. Why the authors fail to use data entry software so that they can minimize data entry errors. Any explanation?

Response: We thank the reviewer for this suggestion, we have added reference for data collection tools (6,14, 30-32). And We appreciate the reviewer's concern regarding the importance of minimizing data entry errors, to minimize data entry errors, we implemented a rigorous data management process. This included double-checking data entries by two independent researchers, conducting thorough data cleaning procedures, and performing consistency checks within SPSS. Additionally, we cross-verified the entered data with the original data sources to ensure accuracy.

Comment

• In the data analysis plan, please mention that the crude and adjusted measures of effect (COR and AOR), how do the authors select candidate variables from bivariate to multivariable model? Did the authors include those variables which were statistically non-significant but clinically important in their model?

• Is it possible to calculate percent if the denominator is less than 100, e.g., the total number of cases is 92, which is <100. Is it appropriate say “Male newborns accounted for 78.3% of cases”?? Please mention only the numbers if the denominator is less than 100. Do not calculate percentages.

• What is the interpretation for the significant difference in the baseline sociodemographic characteristics of the cases and controls? Similarly, the cases and controls were not comparable based on their Obstetric, anthropometrical, and health-related characteristics? If they were not comparable from the beginning, do you think the findings are reliable and generalizable?

• Predictors of Macrosomia: This is not appropriate ate epidemiologic term for your study design. Please revise and present appropriately.

• No need to present the multicollinearity, the omnibus test of model coefficients was significant, Cox and Snell residuals, Nagelkerke’s R2 and the ROC curve analysis in the result section.

• However, the Hosmer and Lemeshow test results should be presented in the final regression table.

• The final regression table should be presented again containing the non-significant variables too. Additionally present the values of for the macrocosmic and normal weight babies cross tab.

Response: We thank again for the above comments. We selected variables with p<0.25 during variable analysis or for crude odds ratio. We have removed omnibus test, residuals result, R squared and ROC curve result. We have used the term associated factors instead of factors to become consistent. Hosmer-lemeshow test is already available in the narration for the regression table. We have added some non-significant sociodemographic variables like residence, educational status, and type of pregnancy.

In the final regression table, there is no need to add cross-tabulations. Because, we already cross-tabulated every categorical and continuous variable with the dependent variable group (case (Macrosomia) or control (Normal weight). So, it is redundancy to add cross-tabulation in the final regression table.

Comment: Additionally, present Asterix or any other symbol for the significant variables.

Response: Thanks for suggestion we have added Asterix on manuscript

Comment: The discussion section is very shallow and try to discuss in detail the possible justification for any similarity or differences in the findings, implication of the findings for community, patients, researchers, and policy makers.

• Try to compare studies conducted in low income set up and specifically Ethiopia and Tigray region.

Response: Thank you for your insightful feedback. We have added a paragraph comparing our findings with those of similar studies conducted in Ethiopia. We believe these additions strengthen the discussion by providing a broader context for our findings and underscoring their relevance to public health in Ethiopia.

Reviewer #2

General Comments:

While the study provides valuable insights into macrosomia in Mekelle City, a more comprehensive review of existing literature on macrosomia in Ethiopia is crucial. The current discussion primarily compares with studies conducted in other countries (China, Nigeria, etc.).

Response: We agree with the reviewer that abstract and introduction section. Could be improved. We have revised this section, we have added a paragraphs to clarify the problem at hand and to establish a strong rational on introduction section

Comments: Please format your abstract to include the following subheadings are Background, Methods, Results, and Conclusions. Therefore, please remove the Objectives subheading and include the text in one of the above sections.

Response: Thank for suggestion, we have formatted abstract section

Comment: The authors should incorporate citations (reference) in some sentences e.g. introduction part paragraph 1 line 2, paragraph 2 lines 2-3.

- Authors should revise paragraphs one and two of the introduction to reduce redundancy. These two sentences convey essentially the same information. To reduce redundancy, the authors could combine the sentences

- In the introduction, the authors should include a dedicated paragraph summarizing key findings from relevant studies conducted in Ethiopia on the prevalence and risk factors of macrosomia

Response: We agree with the reviewer that introduction section. Could be improved. We have revised this section, we have added paragraphs to clarify the problem at hand and to establish a strong rational on introduction section

Comments; Inconsistency in how birth weight is categorized in introduction and methods. If there is a valid reason for using two categories (normal and macrosomia) in the methods and analysis, clearly explain this rationale in the introduction or discussion.

Response: I appreciate you bringing this inconsistency to my attention. To provide readers a comprehensive idea of the subject, the introduction gives a summary of the various birth weight categories, such as low birth weight, normal birth weight, and macrosomia. However, in order to support the study's main goal of identifying predictors of macrosomia, we exclusively focused on two groups in our methodologies and analysis: normal birth weight and macrosomia. This binary classification improves the clarity of understanding results pertaining to macrosomia as the outcome of interest and streamlines statistical analysis.

Comment: - How did the authors screen for pre-existing and gestational diabetes?

Response: Pre-existing diabetes was identified based on documented history of diabetes mellitus (DM) prior to pregnancy, while gestational diabetes was identified from any recorded diagnosis during antenatal care. However, due to the retrospective nature of the study, systematic screening or laboratory confirmation was not performed. This limitation has been acknowledged in the revised manuscript.

Result section---the sentences “Our study indicated that the mean (SD) age of cases and controls was 33.9(4.7) and 27.5(4.0) years. Maternal age or newborn age represent?

Response: It is the age of the mother and we have modified the sentence to add clarity. Thanks for the comment.

Comment- The discussion section should be expanded to include a comparative analysis of the study's findings with those of other studies conducted in Ethiopia. This could highlight similarities, differences, and potential explanations for any observed discrepancies.

Response: Thank you for your insightful feedback. We have added a paragraph comparing our findings with those of similar studies conducted in Ethiopia. We believe these additions strengthen the discussion by providing a broader context for our findings and underscoring their relevance to public health in Ethiopia.

Comment- The authors should discuss the implications of their findings for public health interventions in Ethiopia, considering the specific context and existing healthcare resources.

- Limitation of the study should be discussed, especially the control for preexisting and gestational diabetes which are the very important risk factors for macrosomia

- The authors should also discuss about the generalizability of the study results since the samples were only from mekelle city.

Response: We thank the reviewer for highlighting this important point. While we included history of diabetes mellitus (DM) and family history of DM as variables in our analysis, we acknowledge that these measures do not fully account for the potential impact of preexisting or gestational diabetes on macrosomia. we address this limitation explicitly in the revised manuscript.

---

## [Decision Letter · Decision Letter 1]

14 May 2025

Factors Associated with Macrosomia in Public Hospitals of Mekelle City, Northern Ethiopia: A Multi-Center Study

PONE-D-24-60087R1

Dear Mohamedawel,

We’re pleased to inform you that your manuscript has been judged scientifically suitable for publication and will be formally accepted for publication once it meets all outstanding technical requirements.

Kind regards,

Tamirat Getachew

Academic Editor

PLOS ONE

Additional Editor Comments (optional):

Reviewers' comments:

Reviewer's Responses to Questions

**Comments to the Author**

1. If the authors have adequately addressed your comments raised in a previous round of review and you feel that this manuscript is now acceptable for publication, you may indicate that here to bypass the “Comments to the Author” section, enter your conflict of interest statement in the “Confidential to Editor” section, and submit your "Accept" recommendation.

Reviewer #2: All comments have been addressed

2. Is the manuscript technically sound, and do the data support the conclusions?

Reviewer #2: Yes

3. Has the statistical analysis been performed appropriately and rigorously? 

Reviewer #2: Yes

4. Have the authors made all data underlying the findings in their manuscript fully available?

Reviewer #2: Yes

5. Is the manuscript presented in an intelligible fashion and written in standard English?

Reviewer #2: Yes

6. Review Comments to the Author

Reviewer #2: Thank you for addressing my previous comments. I wish you continued success in your work for the benefit of the wider academic community and beyond.

7. PLOS authors have the option to publish the peer review history of their article (what does this mean? ). If published, this will include your full peer review and any attached files.

**Do you want your identity to be public for this peer review?** For information about this choice, including consent withdrawal, please see our Privacy Policy .

Reviewer #2: **Yes: ** Ebisa Zerihun

---

## [Editor Report · Acceptance letter]

PONE-D-24-60087R1

PLOS ONE

Dear Dr. Ebrahim,

I'm pleased to inform you that your manuscript has been deemed suitable for publication in PLOS ONE. Congratulations! Your manuscript is now being handed over to our production team.

Kind regards,

on behalf of

Dr. Tamirat Getachew

Academic Editor

PLOS ONE